

# The complete chloroplast genome of *Microcycas calocoma* (Miq.) A. DC. (Zamiaceae, Cycadales) and evolution in Cycadales

Aimee Caye G. Chang[1,2,3], Qiang Lai[1], Tao Chen[2], Tieyao Tu[1], Yunhua Wang[2], Esperanza Maribel G. Agoo[4], Jun Duan[1] and Nan Li[2]

[1] South China Botanical Garden, Chinese Academy of Sciences, Guangzhou, China
[2] Shenzhen Fairy Lake Botanical Garden, Chinese Academy of Sciences, Shenzhen, China
[3] University of Chinese Academy of Sciences, Beijing, China
[4] Department of Biology, De La Salle University, Manila, Philippines

## ABSTRACT

Cycadales is an extant group of seed plants occurring in subtropical and tropical regions comprising putatively three families and 10 genera. At least one complete plastid genome sequence has been reported for all of the 10 genera except *Microcycas*, making it an ideal plant group to conduct comprehensive plastome comparisons at the genus level. This article reports for the first time the plastid genome of *Microcycas calocoma*. The plastid genome has a length of 165,688 bp with 134 annotated genes including 86 protein-coding genes, 47 non-coding RNA genes (39 tRNA and eight rRNA) and one pseudogene. Using global sequence variation analysis, the results showed that all cycad genomes share highly similar genomic profiles indicating significant slow evolution and little variation. However, identity matrices coinciding with the inverted repeat regions showed fewer similarities indicating that higher polymorphic events occur at those sites. Conserved non-coding regions also appear to be more divergent whereas variations in the exons were less discernible indicating that the latter comprises more conserved sequences. Phylogenetic analysis using 81 concatenated protein-coding genes of chloroplast (cp) genomes, obtained using maximum likelihood and Bayesian inference with high support values (>70% ML and = 1.0 BPP), confirms that *Microcycas* is closest to *Zamia* and forms a monophyletic clade with *Ceratozamia* and *Stangeria*. While *Stangeria* joined the Neotropical cycads *Ceratozamia*, *Zamia* and *Microcyas*, *Bowenia* grouped with the Southern Hemisphere cycads *Encephalartos*, *Lepidozamia* and *Macrozamia*. All *Cycas* species formed a distinct clade separated from the other genera. *Dioon*, on the other hand, was outlying from the rest of Zamiaceae encompassing two major clades—the Southern Hemisphere cycads and the Neotropical cycads. Analysis of the whole cp genomes in phylogeny also supports that the previously recognized family—Stangeriaceae—which contained *Bowenia* and *Stangeria*, is not monophyletic. Thus, the cp genome topology obtained in our study is congruent with other molecular phylogenies recognizing only a two-family classification (Cycadaceae and Zamiaceae) within extant Cycadales.

Corresponding authors
Jun Duan, duanj@scib.ac.cn
Nan Li, andreali1997@126.com

## INTRODUCTION

Chloroplasts are essential organelles for photosynthesis in plant tissues. Their primary function is to fix carbon by harnessing solar energy to synthesize carbohydrates, pigments and amino acids (*Hong et al., 2017*; *Liu et al., 2017*). These plant organelles, alongside mitochondria, contain their own DNA and a unique set of genes and are also referred to as plastomes (*Wu & Chaw, 2015*) although the bulk of gene content is in the nucleus of most eukaryotic organisms (*Lodish et al., 2000*). Generally, the chloroplast (cp) consists of a large single copy (LSC) region and a small single copy (SSC) region located directly across from each other. Between these two regions are two inverted repeat (IR) sequences (IR$_A$ and IR$_B$). These four regions make up a single quadripartite circular structure about 120–160 kb in length (*Liu et al., 2017*; *Lee et al., 2006*). However, some reports and more recent data from over 1,500 complete genome sequences of chloroplasts from land plants showed that their size ranged up to 170 kb (*Smith, 2017*; *Shaw et al., 2007*).

Chloroplast genomes prove to be advantageous over nuclear genomes in ease of handling and analyzing due to their reduced complexity in structure and dimensions. Despite the relatively smaller genome size, a large amount of data can be obtained using cp genomes such as nucleotide substitutions, pseudogenes, intron losses and variation in genome size. Compared to nuclear and mitochondrial DNA, cp genomic regions are more stable and conserved due to their maternal lineage (*Zhong et al., 2011*) and lack of recombination events (*Martin et al., 2005*). Information gathered from cp genome sequences are also extensively used in addressing uncertainties in evolutionary relationships in various taxa, reconstructing phylogenetic relationships, and gaining a deeper understanding of biogeographical, structural and functional diversity across all organisms (*Jiang, Hinsinger & Strijk, 2016*). These features of chloroplast or plastid DNA (cpDNA or ptDNA) make them ideal models for genome-wide evolutionary studies (*Asaf et al., 2016*; *Choi, Chung & Park, 2016*; *Jiang, Hinsinger & Strijk, 2016*).

Cycads (Cycadophyta) are one particular plant group where studies of molecular phylogeny have been broadly applied (*Liu et al., 2018*; *Gorelick et al., 2014*; *Lu et al., 2014*; *Salas-Leiva et al., 2013*; *Chaw et al., 2005*; *Hill et al., 2003*). Because of their ancestry and close morphological resemblance to some pteridophytes and angiosperms (*Xiao & Moller, 2015*), cycads belong to the earliest lineage among the five major groups of seed plants (Cycadales, Coniferales, Ginkgoales, Gnetales and angiosperms) that date back as far as the Carboniferous Period about 300 Ma (*Jiang, Hinsinger & Strijk, 2016*; *Wu & Chaw, 2015*, *Wu et al., 2007*). Evidences were plant fossils discovered from late Permian determined to be morphologically-related to extant cycads (*Gao & Thomas, 1989*). However, *Nagalingum et al. (2011)* considered that species diversification in cycads occurred recently by analyzing PHYTOCHROME P (PHYP) nuclear gene, in which it showed that the age of extant cycads dates back to only 12 million years. But regardless of

various hypotheses, cycads as a key node in the evolutionary phylogeny of green living things remain true.

Cycads are the second largest group of gymnosperms. They comprise putatively three families, 10 genera and 356 extant species (*Calonje, Stevenson & Osborne, 2019*; *Feng et al., 2017*; *Jiang, Hinsinger & Strijk, 2016*). Currently, assessment on the status of cycads based on IUCN Red List reports that 19.8% are near threatened, 23.1% are vulnerable, 20.4% are endangered, 17.3% are critically endangered and 1.2% are already extinct in the wild (*IUCN, 2019*). Owing to their esthetic nature, cycads are significant in horticulture (*Jiang, Hinsinger & Strijk, 2016*; *Wu & Chaw, 2015*). Unfortunately, exploitation for personal and ornamental use, their slow reproductive rate and habitat destruction have resulted in significant declines in cycad populations and poses a threat to the longevity of the plants and to seed production (*Pinares et al., 2009*; *Stevenson, Vovides & Chemnick, 2003*). One species currently on the verge of extinction is *Microcycas calocoma* (Miq.) A. DC., a species endemic to western Cuba (north to central of Pinar del Rio Province). It grows in montane forests and on limestone in lowlands and grasslands between 50 and 250 m and is currently listed as critically endangered (*IUCN, 2019*; *Bösenberg, 2010*). At present, the monotypic *M. calocoma* is represented by less than 1,000 individuals worldwide. It was previously reported that the female cones are becoming non-receptive to fertilization causing failure in producing viable seeds (*Whitelock, 2002*). However, this claim still requires research for evidences and validation. Most probable factor to unsuccessful fertilization might be due to the limited cycad-pollinator interactions. *Pharaxonotha* is a taxon of beetles known to pollinate New World Zamiaceae such as *Zamia*, *Dioon* and *Ceratozamia*, wherein the species *Papilio esperanzae* was first described in Cuba as potential pollinator of *M. calocoma* found breeding in male cones and feeds on pollen (*Chaves & Genaro, 2005*). According to *Chaves & Genaro (2005)*, this discovery paves a way to investigate factors that influence natural reproduction in *Microcycas* that could also address why some populations fail to reproduce. Other threats commonly encountered by cycads species could be factors induced by humans (e.g., intentional private use, over-collection, habitat destruction, grazing, fire, etc.), by climate (e.g., drought, floods, etc.) and by pressures linked to the biology of cycads some of which were already mentioned above such as reproductive failure, dispersal ability and limited number of identified pollinators to facilitate conservation, among others (*IUCN, 2019*; *Mankga & Yessoufou, 2017*; *Chaves & Genaro, 2005*).

Of the 10 accepted genera in Cycadales, only *Microcycas* lacks a completely sequenced chloroplast genome. This study therefore aimed to determine for the first time the whole chloroplast genome of *M. calocoma* to elucidate the variation and evolution in chloroplast genomes in Cycadales.

## MATERIALS AND METHODS

### DNA extraction, library preparation and sequencing

Five grams of fresh leaves of *M. calocoma* was collected from a plant cultivated in the China National Cycad Conservation Center at Shenzhen Fairy Lake Botanical Garden (Chen T. 2017011701, SZG) for cpDNA isolation using a modified extraction method of

*McPherson et al. (2013)*. After DNA isolation, one μg of purified DNA was fragmented to build short-insert libraries with an insert size of 350 bp following the manufacturer's instructions (Illumina). The short fragments were then sequenced using an Illumina Hiseq 4000 sequencing system at Total Genomics Solution (TGS) in Shenzhen, China (*Borgstrom, Lundin & Lundeberg, 2011*).

## Genome assembly

Approximately 1,538 Mb of raw reads with an average read length of 140 bp was generated from paired-end sequencing. Raw sequence reads were then filtered prior to genome assembly to remove adaptors sequences and 1,244 Mb of cleaned data was used to obtain 165,688 bp assembled chloroplast genome. The cp genome of *Microcycas* was reconstructed by combining de novo and reference-guided assemblies (*Cronn et al., 2008*). The following steps were performed: (1) clean sequence reads were assembled into contigs using SOAPdenovo v2.04 (*Li et al., 2010*) (https://sourceforge.net/projects/soapdenovo2/) short sequence assembly software, then the assembly results were constructed and optimized according to the paired-end and overlapping reads by filling the gaps and removing redundant sequence segments using GapCloser v1.12 (http://soap.genomics.org.cn/soapdenovo.html); (2) for additional guidance in the assembly, BLAST was used to align the contigs to the reference genome *Bowenia serrulata* (GenBank ID: NC_026036.1), which was selected for having the closest sequence length (165,695 bp) with *Microcycas* among available non-Cycadaceae complete chloroplast genome, then aligned contigs (>80% similarity and query coverage) were organized according to the reference genome; and (3) obtained reads were mapped to the assembled draft cp genomes that were visualized by OrganellarGenomeDRAW v1.2 (*Lohse, Drechsel & Bock, 2007*).

## Genome annotation and sequence analysis

Using default parameters to predict protein-coding genes, transfer RNA (tRNA) genes, and ribosome RNA (rRNA) genes, annotation of the chloroplast genes was performed using online processing tools, DOGMA (*Wyman, Jansen & Boore, 2004*) and CpGAVAS software (http://www.herbalgenomics.org/cpgavas/). A whole chloroplast genome BLAST (*Altschul et al., 1990*) search ($E$-value ≤ 1e−5, minimal alignment length percentage ≥40%) was performed against 5 databases: KEGG (Kyoto Encyclopedia of Genes and Genomes) (*Kanehisa et al., 2004*, *2006*; *Kanehisa, 1997*), COG (Clusters of Orthologous Groups), NR (Non-Redundant Protein Database databases) (*Tatusov et al., 2003*; *Tatusov, Koonin & Lipman, 1997*), Swiss-Prot (*Magrane, 2011*), and GO (Gene Ontology) (*Ashburner et al., 2000*) to check and combine annotated genes with their amino acid sequences and corresponding functional annotation information. Finally, the circular *M. calocoma* chloroplast genome map was drawn using OGDraw v1.2 (*Lohse, Drechsel & Bock, 2007*). The sequence was then deposited and can be viewed in National Center for Biotechnology Information (NCBI) database (http://www.ncbi.nlm.nih.gov/nuccore) with accession number MN577566.

## Sequence alignment and phylogenetic analyses

Complete chloroplast sequences available in the NCBI database were retrieved and included in the dataset for phylogenetic analyses. All currently available sequenced whole cp genomes under Cycadales and Ginkgoales were obtained. A total of 15 sequences were included in the data set (Table S1).

Eighty-one protein-coding genes encompassing LSC, SSC and IR regions and excluding intergenic spacers were extracted from the plastid genomes followed by alignment of each gene using MAFFT multiple aligner with default parameters in Geneious Prime v2019.0.3 (https://www.geneious.com, *Kearse et al., 2012*). Ambiguous regions that have low nucleotide similarities causing large alignment gaps were then manually removed from some genes, after which, all aligned genes were concatenated resulting in total length of 55,971 bp prior to phylogenetic analyses. The data matrix was subjected to maximum likelihood (ML) phylogenetic reconstruction using RAxML v.8 (*Stamatakis, 2014*) with GTR + I + G substitution model selected by jmodeltest2 and 1,000 bootstrap replicates. Bayesian inference (BI) was also calculated using MrBayes 3.2.6 using a GTR model of nucleotide substitution and a gamma distribution rate variation across sites (*Ronquist & Huelsenbeck, 2003*) setting four MCMC running for one million generations with sampling every 1,000 generations and the first 25% discarded as burn-in. Branches with bootstrap values >75 for ML and Bayesian posterior probabilities >0.95 for BI were considered as highly supported. Both ML and BI analyses were conducted in CIPRES Science Gateway (www.phylo.org) (*Miller, Pfeiffer & Schwartz, 2010*). Alignment of sequence identities were plotted using mVISTA online platform (*Frazer et al., 2004*; *Mayor et al., 2000*). A total of 10 genera of Cycadales were included to analyze similarity profiles on coding and noncoding regions across the genomes. Shuffle-LAGAN alignment program was selected as a parameter prior to run the analysis (*Brudno et al., 2003*) as this setting performs global alignment that detects rearrangements and inversions.

## RESULTS

### Plastome size and features of *M. calocoma*

The genome size of the complete chloroplast DNA of *M. calocoma* is 165,688 bp in length with 86 protein-coding genes (88,778 bp), 39 tRNA genes (9,050 bp) and eight ribosomal RNA genes (2,906 bp). In *M. calocoma*, the *tuf*A gene contains five premature stop codons within the sequence therefore making it a pseudogene. One hundred and thirty three genes were organized in a circular quadripartite structure consisting of large and SSC regions separated by two inverted repeat regions (Fig. 1) composed of 90,651 bp, 22,741 bp and a pair of 26,184 bp regions, respectively. In the entire cp genome of *M. calocoma*, four protein-coding (*ndh*B, *rps*7, *rps*12, *ycf*2), four rRNA (*rrn*16, *rrn*23, *rrn*4.5, *rrn*5) and seven tRNA genes (*trn*A-UGC, *trn*H-GUG, *trn*I-GAU, *trn*L-CAA, *trn*N-GUU, *trn*R-ACG, *trn*V-GAC) were duplicated in the IR regions (Table 1). The protein-coding gene *rps*12 was also trans-spliced in the LSC aside from being duplicated in the IRs. The presence of two split genes (*trn*K-UUU/*mat*K and *atp*B/*atp*E) in

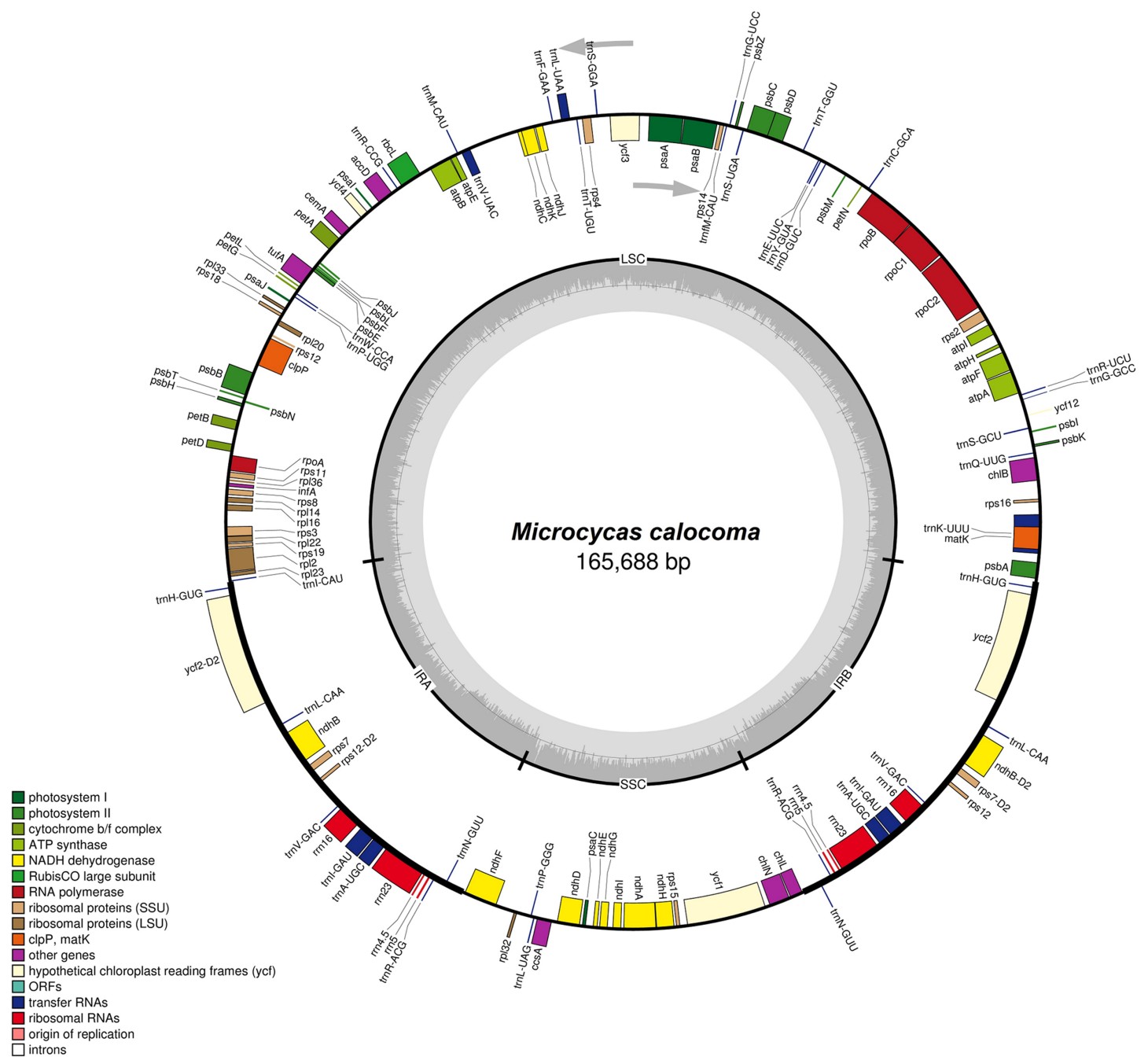

**Figure 1** *Microcycas calocoma* **annotated circular genome.** Gene map of *Microcycas calocoma* whole chloroplast genome. Innermost circle indicates partition of large single copy (LSC) region, small single copy (SSC) region and two regions of inverted repeats (IR$_A$ and IR$_B$) which are also highlighted by the thick lines found in the outermost circle. Genes within the outermost circle are transcribed in clockwise direction whereas genes on the outside are transcribed in counter clockwise direction. Colors denote the different functions of genes.

the LSC were also found. The overall GC content of *M. calocoma* (39.6) falls within the range of sequenced chloroplast genomes of cycads which range from 39.4% to 40.1%. LSC, SSC and IR regions have 38.8%, 37.2% and 41.8% GC content, respectively.

**Table 1  Genes annotated in *Microcycas calocoma*.** List of genes encoded by *Microcycas calocoma*.

| Gene category | Gene names | Database accessions |
|---|---|---|
| ATP Synthase | *atp*A, *atp*B, *atp*E, *atp*F, *atp*H, *atp*I | GO:0005524 |
| | | GO:0009507 |
| | | GO:0015986 |
| | | GO:0045263 |
| | | GO:0009507 |
| | | GO:0016020 |
| NADH dehydrogenase | *ndh*A, *ndh*B$^{(x2)}$, *ndh*C, *ndh*D, *ndh*E, *ndh*F, *ndh*G, *ndh*H, *ndh*I, *ndh*J, *ndh*K | GO:0003954 |
| Cytochrome b/f complex | *pet*A, *pet*B, *pet*D, *pet*G, *pet*L, *pet*N | GO:0009512 |
| Photosystem I | *psa*A, *psa*B, *psa*C, *psa*I, *psa*J | GO:0009522 |
| Photosystem II | *psb*A, *psb*B, *psb*C, *psb*D, *psb*E, *psb*F, *psb*H, *psb*I, *psb*J, *psb*K, *psb*L, *psb*M, *psb*N, *psb*T, *psb*Z | GO:0009523 |
| RubisCO large subunit | *rbc*L | GO:0009507 |
| Ribosomal protein genes (large subunits) | *rpl*14, *rpl*16, *rpl*2, *rpl*20, *rpl*22, *rpl*23, *rpl*32, *rpl*33, *rpl*36 | GO:0015934 |
| Ribosomal protein genes (small subunits) | *rps*11, *rps*12$^{(x3)}$, *rps*14, *rps*15, *rps*16, *rps*18, *rps*19, *rps*2, *rps*3, *rps*4, *rps*7$^{(x2)}$, *rps*7, *rps*8 | GO:0015935 |
| RNA Polymerase | *rpo*A, *rpo*B, *rpo*C1, *rpo*C2 | GO:0009507 |
| | | GO:0003677 |
| | | GO:0009507 |
| | | GO:0009536 |
| Ribosomal RNA genes | *rrn*16$^{(x2)}$, *rrn*23$^{(x2)}$, *rrn*4.5$^{(x2)}$, *rrn*5$^{(x2)}$ | GO:0003735 |
| | | GO:0003743 |
| Transfer RNA genes | *trn*A-UGC$^{(x2)}$, *trn*C-GCA, *trn*D-GUC, *trn*E-UUC, *trn*F-GAA, *trn*fM-CAU, *trn*G-GCC, *trn*G-UCC, *trn*H-GUG$^{(x2)}$, *trn*I-GAU$^{(x2)}$, *trn*K-UUU, *trn*L-CAA$^{(x2)}$, *trn*L-UAG, *trn*M-CAU, *trn*N-GUU$^{(x2)}$, *trn*P-GGG, *trn*P-UGG, *trn*Q-UUG, *trn*R-ACG$^{(x2)}$, *trn*R-CCG, *trn*R-UCU, *trn*S-GCU, *trn*S-GGA, *trn*S-UGA, *trn*T-GGU, *trn*T-UGU, *trn*V-GAC$^{(x2)}$, *trn*V-UAC, *trn*W-CCA, *trn*Y-GUA | GO:0030533 |
| | | GO:0006412 |
| ATP-dependent protease | *clp*P | GO:0008462 |
| Maturase | *mat*K | GO:0009536 |
| Hypothetical chloroplast reading frames | *ycf*1, *ycf*12, *ycf*2$^{(x2)}$, *ycf*3, *ycf*4 | GO:0016021 |
| | | GO:0016021 |
| | | GO:0009507 |
| | | GO:0009507 |
| | | GO:0015979 |
| Acetyl-CoA carboxylase | *acc*D | GO:0009536 |
| C-type cytochrome synthesis gene | *ccs*A | GO:0009507 |
| Envelope membrane protein | *cem*A | GO:0016020 |
| Photochlorophyllide reductase | *chl*B, *chl*L, *chl*N | GO:0016630 |
| Translational initiation factor | *inf*A | GO:0009536 |

## Plastome features in Cycadales

Thirteen completely-sequenced chloroplast genomes, representing nine of the ten genera of cycads, were in GenBank during our study. The addition of the plastid genome of *M. calocoma* makes known the sequences of all genera of Cycadales. Summary of the genome features in all sequenced cp genomes of cycads was shown in (Table 2).

The smallest genome with 161,815 bp is in *Dioon spinolosum* while *Macrozamia mountperriensis* consisting of 166,341 bp has the largest. Despite significant difference in

**Table 2 Gene features of *Microcycas calocoma*.** Characteristics of complete chloroplast genomes of Cycadales showing genome lengths, gene numbers and GC content. Data are retrieved from GenBank and accession numbers are indicated.

| Species (Accession No.) | Genome size (bp) | LSC (bp) | SSC (bp) | IRs (bp) | Number of genes | | | | | GC content (%) |
|---|---|---|---|---|---|---|---|---|---|---|
| | | | | | Total | Coding | tRNA | rRNA | Pseudo-genes | |
| *Microcycas calocoma* (MN577566) | 165,688 | 90,651 | 22,741 | 26,148 | 134 | 86 | 39 | 8 | 1 | 39.6 |
| *Cycas revoluta* (NC_020319.1) | 162,489 | 88,977 | 23,376 | 25,068 | 157 | 109 | 39 | 8 | 1 | 39.4 |
| *Cycas debaoensis* (KM459003) | 162,094 | 88,852 | 23,088 | 25,076 | 133 | 87 | 37 | 8 | 1 | 39.4 |
| *Cycas debaoensis* (KU743927) | 162,092 | 88,854 | 23,088 | 25,076 | 133 | 87 | 37 | 8 | 1 | 39.4 |
| *Cycas taitungensis* (NC_009618.1) | 163,403 | 90,216 | 23,039 | 25,074 | 169 | 122 | 38 | 8 | 1 | 39.5 |
| *Cycas panzhihuaensis* (NC_031413.1) | 162,470 | 88,932 | 23,488 | 25,045 | 157 | 109 | 39 | 8 | 1 | 39.4 |
| *Stangeria eriopus* (NC_026041.1) | 163,548 | 89,850 | 23,006 | 25,346 | 134 | 81 | 39 | 8 | 6 | 39.5 |
| *Bowenia serrulata* (NC_026036.1) | 165,695 | 90,733 | 23,156 | 25,903 | 134 | 87 | 38 | 8 | 1 | 39.9 |
| *Dioon spinolosum* (NC_027512.1) | 161,815 | 88,756 | 23,355 | 24,852 | 135 | 87 | 39 | 8 | 1 | 40.1 |
| *Encephalartos lehmannii* (NC_027514.1) | 165,822 | 90,724 | 23,302 | 25,898 | 135 | 87 | 39 | 8 | 1 | 39.9 |
| *Macrozamia mountperriensis* (NC_027511.1) | 166,341 | 91,171 | 23,334 | 25,918 | 135 | 87 | 39 | 8 | 1 | 39.8 |
| *Lepidozamia peroffskyana* (NC_027513.1) | 165,939 | 90,804 | 23,299 | 25,918 | 135 | 87 | 39 | 8 | 1 | 39.9 |
| *Ceratozamia hildae* (NC_026037.1) | 165,733 | 90,487 | 22,973 | 26,137 | 135 | 87 | 39 | 8 | 1 | 39.7 |
| *Zamia furfuracea* (NC_026040.1) | 164,953 | 90,441 | 23,228 | 25,642 | 135 | 87 | 39 | 8 | 1 | 39.7 |

genome length, both species contain 135 genes (87 protein-coding, 39 tRNA, eight rRNA, one pseudogene) indicating that an increase in plastome size does not denote addition of genes or functions. The number of genes in cycads typically ranges from 133 to 135 except in *C. revoluta* and *C. panzhihuaensis*, both of which have 157 genes, and *C. taitungensis* which has 169 genes, the highest number in cycads. These three species of *Cycas* also have the highest number of protein-coding genes among cycads with 109 genes in *C. revoluta* and *C. panzhihuaensis* and 122 genes in *C. taitungensis*. Variation in the number of tRNAs was also observed. *Cycas debaoensis* has 37, *C. taitungensis* and *B. serrulata* both have 38, while the remaining cycads have 39 tRNAs. In comparison, the number of ribosomal RNA (rRNA), eight, is the same in all cycads.

The majority of cycads have 87 protein-coding genes and one pseudo-*tuf*A gene. Interestingly, *Stangeria eriopus* has 81 coding genes and six reported pseudogenes–*chl*L, *chl*N, *rpl*23, *tuf*A, *chl*B and *mat*K genes (*Wu & Chaw, 2015*). *M. calocoma* contains one

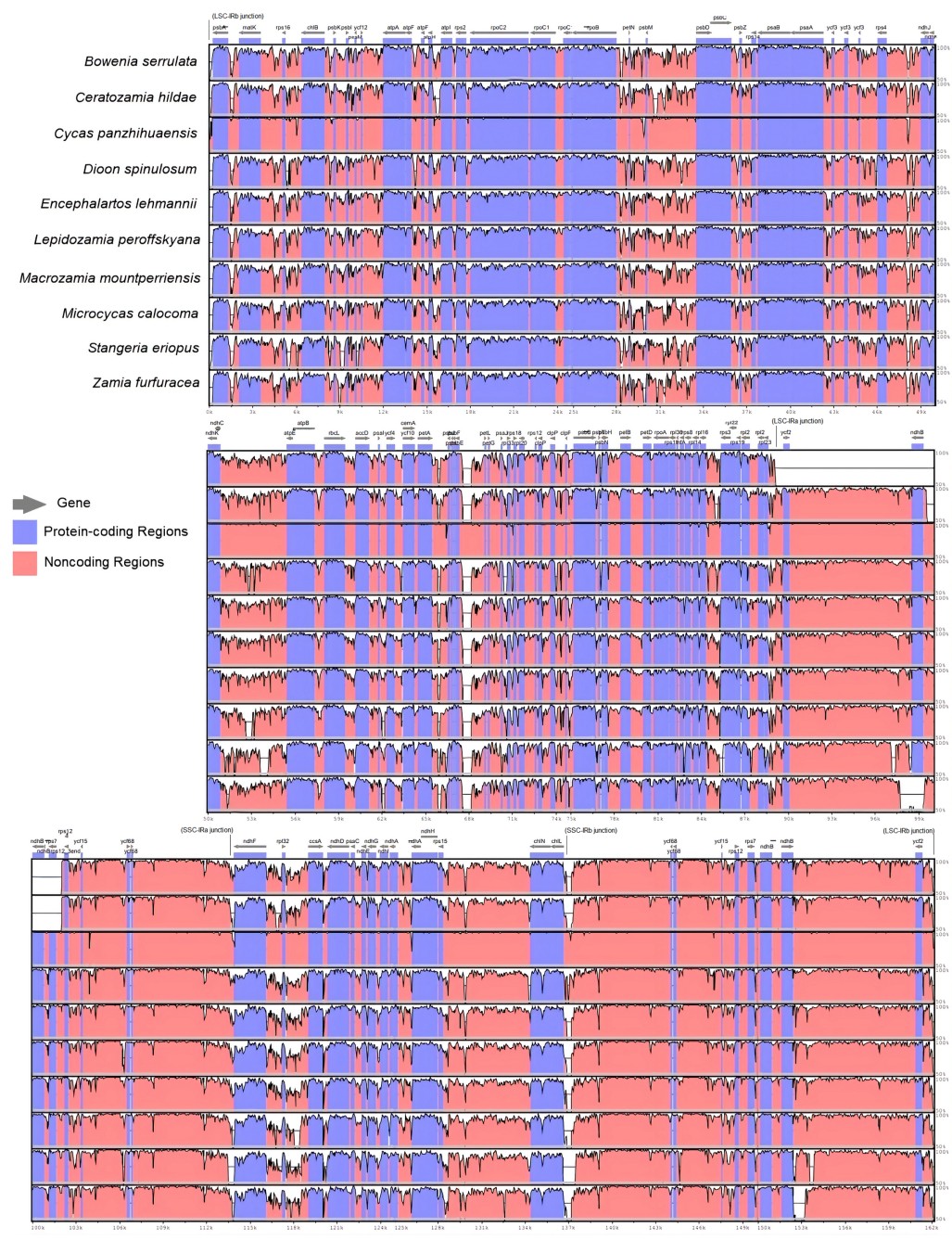

**Figure 2 Sequence identity plots of various genera in Cycadales using mVISTA.** Sequence variations among 10 genera in Cycadales using *C. debaoensis* as base reference. Complete chloroplast genomes generated by mVISTA. Regions in pink indicate conserved non-coding sequences, purple are conserved exons, while white-colored regions identify more variable sites. The *Y*-axis represents percent identity ranging from 50% to 100%. IR junctions are indicated in parentheses to show LSC, SSC and IR regions.

*tuf*A pseudogene (Ψ*tuf*A) which is also in all cycad genomes. *Jiang, Hinsinger & Strijk (2016)* reported that the Ψ*tuf*A gene of *C. debaoensis* was one base pair longer than in *C. taitungensis* (723 bp). In *M. calocoma*, this pseudogene is 1,233 bp in length.

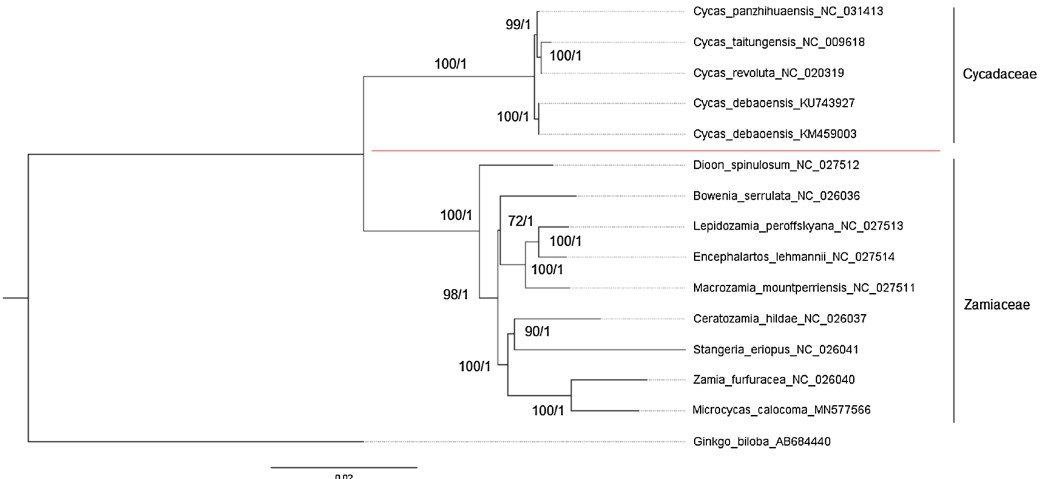

**Figure 3 cpDNA tree obtained using ML and BI.** Maximum likelihood (ML) tree and Bayesian inference (BI) tree of concatenated data matrix of 81 protein-coding genes from complete chloroplast genome sequences of 10 genera of Cycadales available in GenBank. *Ginkgo biloba* was used as outgroup. Bootstrap support values are given at the nodes annotated as ML/BI. Branches with bootstrap values >75 for ML and Bayesian posterior probabilities >0.95 for BI were considered as highly supported.

## Sequence variation in cycads

Using one representative per genus, mVISTA produced a plot showing peaks and valleys where pink regions denote conserved non-coding sequences (CNS). White regions indicate sites within the genome that are indistinctly aligned, suggesting more variable sites. These regions happen to coincide with the inverted repeat regions of all chloroplast genomes suggesting that IRs are prone to polymorphisms in comparison with the LSC and SSC regions. High degree of synteny in the coding regions was observed across all genomes of Cycadales when *C. debaoensis* was used as a base reference, as indicated by the purple-colored blocks (conserved exons) for all genomes (Fig. 2). Due to the lack of other representative complete chloroplast genomes under Zamiaceae, a *Cycas* species was used as reference. Since a *Cycas* species was used as base reference, only *C. panzhihuaensis* showed an obviously unique pattern indicating that it shares more similar regions with the reference as they are more closely related, setting it apart from the identity plots of the rest of the species which all belong to Zamiaceae. Divergence in CNS regions was higher whereas variations in coding regions were less discernible. Most divergent regions were apparent in the intergenic spacers among all cp genomes. As shown in our alignment, these highly varied sites include regions between *psbA*-*matK*, *petA*-*psbJ*, *petL*-*psbE*, *ndhF*-*ccsA*, among others. Nonetheless, these plots suggest high similarities in the nucleotide sequences within Cycadales. It also denotes that slow evolution occurs in Cycadales where gene order is highly conserved and a reduced amount of gene rearrangements occur.

## Phylogenetic analyses

The phylogeny obtained from the complete chloroplast sequences of cycads using ML and BI revealed several consistencies from previous studies employing various gene segments

from nuclear and/or chloroplast DNA (Fig. 3). Both trees obtained showed significant support (>70 for ML and >0.95 Bayesian Posterior Probability for BI) and consistency in tree topologies. These include the monophyly of the three major clades: (1) all *Cycas* spp., (2) the Encephalartoid clade or "Southern Hemisphere clade" consisting of *Encephalartos-Lepidozamia-Macrozamia* genera, and (3) the "Neotropical clade" comprising of *Ceratozamia-Microcycas-Zamia* group. Moreover, *Dioon* was confirmed as sister to the rest of Zamiineae (*Liu et al., 2018*; *Lu et al., 2014*; *Gorelick et al., 2014*; *Salas-Leiva et al., 2013*).

Phylogenetic analyses also showed that *Microcycas* is closer to *Zamia* than to other genera of cycads which is congruent with previous studies using chloroplast *matK* and *rbcL* genes, *trnK* intron, nuclear rDNA ITS gene and concatenated nuclear and plastid DNA sequences (*Lu et al., 2014*; *Salas-Leiva et al., 2013*; *Chaw et al., 2005*; *Hill et al., 2003*). Furthermore, the monogeneric Cycadaceae forms the basal-most clade wherein *C. panzhihuaensis* and the sisters *C. taitungensis* and *C. revoluta* diverged from *C. debaoensis*. The phylogenetic placements of *Bowenia* and *Stangeria* were controversial, where older references recognize that they belong to the same family Stangeriaceae (*Stevenson, 1992*; *Stevenson, 1981*). In our study, these two genera separated into different clades. Even though showing weaker ML support, *Bowenia* (72 ML and 1.0 BPP) grouped with the Encephalartoid clade while *Stangeria*, with significant support (90 ML and 1.0 BPP), was sister to *Ceratozamia* clustered in the Neotropical clade. As the results suggest that the African cycad *Stangeria* is closest to Neotropical cycads instead of its supposed closest relatives in terms of morphology (*Bowenia*) and biogeographic location (*Encephalartos*), this interesting classification will be further discussed below, as well as the placement of *Bowenia* with the Southern Hemisphere cycads.

## DISCUSSION

### Plastomes of cycads

Based on previously-completed sequences, 100–130 unique genes are contained within chloroplast genomes of cycads (*Jiang, Hinsinger & Strijk, 2016*; *Wu & Chaw, 2015*; *Wu et al., 2007*). Approximately 79 of these genes code for proteins involved in metabolic pathways for photosynthesis whereas the remaining 30 are responsible for coding transfer RNAs (tRNAs) and four are responsible for ribosomal RNAs (rRNAs) (*Hong et al., 2017*; *Dong et al., 2013*; *Yang et al., 2013*). In this study, the whole chloroplast genome of *Microcycas calcoma* was presented and 13 complete chloroplast genomes of cycads representing nine genera within Cycadales were used for comparison. Hence, all the ten genera within Cycadales now have at least one species with complete chloroplast genome.

The gene organization, content and order in cycad genomes appear to be highly conserved in all ten genera. This agrees with previous reports (*Rai et al., 2003*; *Wu et al., 2007*) and supports the slow evolution of cycads in which the changes that occur in the genomes are very minimal since they diverged in the early Jurassic (*Nagalingum et al., 2011*; *Wu & Chaw, 2015*). Our study showed that even the universal barcode loci *matK* and *rbcL* are not highly variable in cycads. On the contrary, we observed that *ycf*1, *ycf*2 and *accD* genes are highly polymorphic sites across all cycad plastomes.
The nonfunctional gene, tufA, was inherited from the charophyte lineage of green algae that once coded for a functional protein elongation factor (*Wu et al., 2007*). Functional *tuf*A genes are present in the cpDNA of algae and bacteria; and are also known to be transferred from the cpDNA to the nuclear ribosomal DNA in rice, legumes, tobacco and *Arabidopsis* (*Wu et al., 2007*; *Sugita, Murayama & Sugiura, 1994*; *Baldauf & Palmer, 1990*). Pseudo-*tufA* genes are also in the genomes of *Anthoceros* (hornwort) and *Ginkgo*. In cycads, the highest frequency of pseudonized genes (6) were present in *S. eriopus* (*Wu et al., 2007*).

## Sequence variations and evolution in Cycadales

Sequence identity plots of cycad genomes identified highly conserved coding (purple) regions. Non-coding (pink) regions appeared more divergent whereas white areas mostly coinciding with IR regions also showed less sequence similarities. According to *Wu & Chaw (2015)*, gene conversions frequently occur in IRs, which favors a GC-biased (gBGC) substitution mechanism. gBGC is believed to be the mechanism employed by cycads in amending plastome-wide mutations, therefore playing a role in evolutionary stasis and genome stability of chloroplast genomes (*Wu & Chaw, 2015*). The results of the global sequence alignment agreed with the phylogeny employing 81 protein-coding genes, obtained using ML and BI with high support values (>70% and = 1.0 BPP). It showed that the three major clades that grouped together had common identity profiles, thus denoting close relationships. These three are the *Cycas* group, the Australian genus *Bowenia* joining the Southern Hemisphere group (*Encephalartos-Lepidozamia-Macrozamia*), and South African genus *Stangeria* joining the Neotropical group (*Ceratozamia-Microcycas-Zamia*). Geographical patterns and distribution of cycads coincide with our phylogenetic analyses using cp genomes but with few exceptions discussed in the succeeding sections such as the case of African genera *Stangeria* and *Encephalartos* which were separated into different clades.

The use of whole chloroplast genomes shows a clear delineation separating Cycadaceae from the rest of the cycads. But for Zamiaceae, numerous studies have shown that the taxonomic placement of *Bowenia*, *Stangeria* and *Dioon* was ambiguous and their phylogeny remains unresolved (*Gorelick et al., 2014*; *Bogler & Francisco-Ortega, 2004*). In this study, *Dioon* was basal among the three genera, *Bowenia* formed a monophyletic group with *Encephalartos-Lepidozamia-Macrozamia* and *Stangeria* was sister to *Ceratozamia* forming a monophyletic clade with *Microcycas* and *Zamia*. This is congruent with other phylogenies wherein *Dioon* separated from other genera and contained two major clades (*Salas-Leiva et al., 2013*; *Bogler & Francisco-Ortega, 2004*; *Hill et al., 2003*). Morphological synapomorphies that can be correlated in our cpDNA topology include the presence of stomata on the sporangia of all Zamiaceae cycad species (*Dehgan, Schutzman & Almira, 1993*); *Encephalartos*, *Lepidozamia* and *Macrozamia* all have lateral lobes in the megasporophylls (*Stevenson, 1990*); *Microcycas* and *Zamia* have asymmetrically-branched trichomes (*Stevenson, 1990*); *Encephalartos* and *Lepidozamia* have a common mucilage chemistry not found in other cycads (*De Luca et al., 1982*); and the Australian *Macrozamia* and African *Encephalartos* both having pith structures intersected by gum canals and irregular cauline vascular bundles, in contrast to the

peduncular vascular bundles found in *Dioon*, *Stangeria*, *Ceratozamia* and *Zamia* (*Berry, 1918*). Moreover, *Stangeria* and *Bowenia* share few similarities with some members of their respective clades such as the formation of tracheids in *Stangeria* and *Zamia* are scalariform, and the orientation of vascular bundles in the tertiary cambium of *Bowenia* and *Macrozamia* are both inverted (*Govil, 2014*; *Berry, 1918*).

*Stevenson (1981)* previously described a family Boweniaceae for the genus *Bowenia* but striking morphological similarities eventually unite *Bowenia* and *Stangeria* in Stangeriaceae (*Hill et al., 2003*; *Stevenson, 1992*). These characters similar to both genera are vascularized stipules, amphivasal cotyledon bundles, leaflet traces coming from more than one rachis bundle, uneven production of cataphylls, ovules connected under the megasporophyll stalk, and purple sarcotesta. Although morphological trees group these two genera together, molecular data always show weak affinity of the two suggesting that these characters evolved independently from the two genera (*Hill et al., 2003*; *Bogler & Francisco-Ortega, 2004*). A study by *Griffith et al. (2012)* using combined DNA sequences, morphology and phenology data also did not show evidences of strong connection between these two genera. *Bowenia* formed a distinct clade second to *Dioon* within Zamiaceae whereas *Stangeria* was embedded in a clade with *Ceratozamia*, *Zamia* and *Microcycas* (*Griffith et al., 2012*). The results of our study suggest that *Bowenia* and *Stangeria* are indeed taxonomically distant from each other despite having similar morphological features and being placed previously in the same family Stangeriaceae. The topology obtained in our study therefore supports the formal classification of only two families, Cycadaceae and Zamiaceae, within extant Cycadophyta (*Christenhusz et al., 2011*). The previous placement of *Bowenia* and *Stangeria* under the same family was in fact inaccurate (*Gorelick et al., 2014*; *Salas-Leiva et al., 2013*; *Chaw et al., 2005*; *Bogler & Francisco-Ortega, 2004*; *Hill et al., 2003*) based on molecular data.

Differences in chromosome number and karyotype were also linked to the phylogenetic separation of *Bowenia* ($2n = 18$) and *Stangeria* ($2n = 16$) (*Rastogi & Ohri, 2019*; *Moretti, 1990*). This also provides an explanation to the phylogenetic affinity of *Stangeria* to *Ceratozamia* as they had the most similar karyotypes in Cycadales both having 12 metacentric, two submetacentric and two telocentric chromosomes (*Kokubugata et al., 2001*). Mapping the distribution of 18S and 26S rDNA sites in the somatic chromosomes of both genera also revealed similar dispersed patterns (*Kokubugata, Hill & Kondo, 2002*). In addition, mapping of 5S rDNA site is similarly situated at the interstitial region of two metacentric chromosomes. However in *Ceratozamia*, it was positioned proximal to the center, whereas in *Stangeria*, it was located near the distal end. This indicates that at least one paracentric inversion might have occurred in the past evolution of these two genera (*Kokubugata, Vovides & Kondo, 2004*).

*Salas-Leiva et al. (2013)* were not able to specify morphological synapomorphies that support the relation of *Stangeria* with Neotropical cycads, as well as *Bowenia* with Southern Hemisphere cycads. This led us to search for shared morphological characters that would unite *Stangeria* and *Bowenia* with their respective clades in our whole cpDNA tree. One similarity observed is the obscurity of veins in the leaflets of *Bowenia* and Southern Hemisphere cycads, whereas in the clade comprising *Stangeria* and Neotropical

cycads, the veins tend to be more embossed. However, this needs further validation especially in some large genera such as *Encephalartos* and *Zamia* as they seem to exhibit more diversity in articulation of veins. Another shared characteristic is the type of stem wherein cycads can be divided into two categories. In *Dioon*, *Stangeria*, *Ceratozamia*, *Zamia* and *Microcycas*, the stem is monoxylic characterized by narrow cambiums with only one ring of secondary xylem and no apparent successive growth. Whereas in *Bowenia*, *Encephalartos*, *Lepidozamia* and *Macrozamia*, also including *Cycas*, the stem is polyxylic characterized by multiple layers of cortical cambiums and secondary thickening (*Govil, 2014*; *Berry, 1918*; *Chamberlain, 1912*). Moreover, *Dehgan & Dehgan (1988)* studied the pollen morphology within Cycadales and determined that *Bowenia* resembled a psilate exine surface and have relatively thin walls and similar alveolar arrangement with the pollen of all Southern Hemisphere cycads. On the other hand, *Stangeria* shares a foveolate exine surface and relatively thick walls with all Neotropical cycads. These characteristics can also be attributed to the pollination biology of Cycadales. *Norstog (1987)* hypothesized three pollination modes in cycads: anemophily (pollination by wind), entomophily (pollination by insects), and amphiphily (anemophily succeeded by entomophily). Although all genera are likely to be amphiphilous, taxa whose pollen surface lack ornamentation and walls are relatively thin (*Macrozamia*, *Lepidozamia*, *Encephalartos*, *Bowenia*) are presumably anemophilous as the smooth, non-sticky and thin surface makes the pollen easier to be dispersed by wind (*Dehgan & Dehgan, 1988*; *Norstog, 1987*; *Thanikaimoni, 1986*). On the contrary, cycads whose pollen surface consist of coarse pits and walls are relatively thick (*Ceratozamia*, *Dioon*, *Microcycas*, *Stangeria*, and *Zamia*) are probably entomophilous as the rough and thick surface seem to be more attuned to pollination by insects (*Dehgan & Dehgan, 1988*; *Norstog, 1987*; *Tang, 1987*; *Punt, 1986*). In addition, pollen grain morphology of *Dioon* and *Stangeria* were found to be highly similar having subcircular outline and foveolate exine surface. Aside from having a single vascular ring, both genera also has a common leaflike megasporophyll which was presumed to be a shared primitive feature that was conserved independently (*Dehgan & Dehgan, 1988*).

The African genera *Encephalartos* and *Stangeria* were separated into different clades suggesting that their evolutionary origin may have occurred before South American, African and Australian continents split (*Hill et al., 2003*) that accounts for their shared morphological and molecular features with other genera in Zamiaceae. The morphological similarities discussed in the previous section somehow provide biogeographical linkage between the African *Stangeria* and the cycads endemic to the Americas (*Dioon*, *Ceratozamia*, *Microcycas* and *Zamia*). In our hypothesis, the lack of diverse morphological synapomorphies to resolve taxonomic placements of *Dioon*, *Stangeria* and *Bowenia* might be due to the fact that only a number of extant cycads remain and the lack of these presumed extinct species are causing discrepancies in consolidating biogeographical, morphological and molecular phylogenies. Hence we place an emphasis on the hypothesis that around 200-135 MYA in Gondwana (*Bogler & Francisco-Ortega, 2004*; *Jokat et al., 2003*), the southern part of Pangaea where Africa, the Americas, Antarctica, India and Australia were once connected, there might exist other ancestral cycads species common

between (1) *Dioon-Stangeria*, (2) *Stangeria-Ceratozamia*, (3) *Stangeria-Zamia-Microcycas*, (4) *Stangeria-Bowenia*, and (5) *Encephalartos-Lepidozamia-Macrozamia*. To support this theory, the latter two would therefore require common ancestor/s situated in Antarctica wherein the shift in climate and environment after fragmentation of the land masses caused eventual extinction of the presumed cycad ancestors. This theory is very likely as cycad fossils from the genus *Eostangeria* was already discovered in North America and Europe, believed to have existed before the final division of Pangaea supercontinent (*Uzunova, Palamarev & Kvacek, 2001*; *Kvacek & Manchester, 1999*). A fossil cycad taxa, *Restrepophyllum*, was also discovered in Patagonia, Argentina and found to be morphologically similar to *Zamia* as well as few shared characters with *Stangeria*. This suggests a possible northward migration of *Zamia* from South America and that members of Zamiaceae and putative Stangeriaceae were widespread in Patagonia during the Early Cretaceous (*Passalia, Del Fueyo & Archangelsky, 2010*). Other fossil cycad taxa include genera *Dioonopsis* found in Northeast Japan (*Horiuchi & Kimura, 1987*) and western North America (*Erdei, Manchester & Kvaček, 2012*), *Encephalartites* (*Takimoto & Ohana, 2016*) in Northeast Japan, *Crossozamia* in Taiyuan, China (*Gao & Thomas, 1989*), *Antarcticycas* in Antarctica (*Smoot, Taylor & Delevoryas, 1985*) and *Zamia* fossil species discovered in Panama (*Erdei et al., 2018*). Most of these fossil discoveries were found to be morphologically linked to extant cycads under Zamiaceae, providing plausible explanations to the diverse morphological characters observed in Zamiaceae species compared to the more conserved characters in Cycadaceae. Phylogenetic studies were mostly restricted on extant cycads and these lost ancestors might be the missing links that could bridge the gap on the taxonomic issues of *Dioon* and the affinity of *Stangeria* and *Bowenia* with Neotropical cycads and Southern Hemisphere cycads, respectively, despite their differences in distribution and morphology. Only upon integration of extinct and extant cycads will we gain profound understanding on this intriguing group of plants.

## CONCLUSION

The use of complete chloroplast genome proved to be advantageous in the phylogenetic reconstruction of Cycadales due to its conserved genomic structure and arrangement. Likewise, it holds sufficient genetic information to discriminate variation between species. Here, we present the 165,688 bp sequence of the whole chloroplast genome of *M. calocoma* which is within the range of all previously sequenced cycad genomes. The phylogenetic analysis of Cycadales confirmed that *Microcycas* is closely related to *Zamia* and sister to *Ceratozamia*, and that *Stangeria* and *Bowenia* are in different groups; thus stressing that their previous placement in a single family was definitely inappropriate. Hence, our resulting taxonomic classification using chloroplast genomes are consistent with all other molecular phylogenies of cycads (*Salas-Leiva et al., 2013*; *Christenhusz et al., 2011*; *Bogler & Francisco-Ortega, 2004*; *Hill et al., 2003*).

Furthermore, sequence variation analyses showed that all cycad genomes share highly similar genomic profiles confirming their slow evolution and reduced mutations in comparison with other gymnosperms and seed plants. No rearrangements in the genome structure were detected. This was expected as cycads are known to have a static

evolutionary history, undergoing few changes in morphology since the Jurassic Period (*Wu & Chaw, 2015*; *Nagalingum et al., 2011*). However, the common identity profiles were shared less in the inverted repeat regions as well as non-coding sequence regions, suggesting their potential use as regions of interest in studying cycad evolution and phylogeny. In Zamiaceae, only a single species representing each genus has a completely-sequenced chloroplast genome. Thus, adding more sequences to large genera, such as *Zamia* and *Encephalartos*, is crucial to validate existing phylogenetic data and to bridge knowledge gaps regarding cycads species divergence in land plant evolution. Furthermore, the need to expand our knowledge on the field of paleobotany, pollination biology and anatomical and morphological characterization of cycads opens up research opportunities to address incongruence between morphological and molecular phylogenies in Cycadales.

### Funding
This research was supported by the special financial project from the National Forestry and Grassland Administration for Rescues and Breeding of Wild Rare and Endangered Species (Li Nan 2017, 2018, 2019), the Shenzhen Urban Management Bureau (Li Nan 201411) and the Shenzhen Key Laboratory of Tropical and Subtropical Plant Diversity, Fairy Lake Botanical Garden, Chinese Academy of Sciences (Chen Tao 2017–2019), and South China Botanical Garden (Duan Jun 2017–2019), Chinese Academy of Sciences. There was no additional external funding received for this study. The funders had no role in study design, data collection and analysis, decision to publish, or preparation of the manuscript.

### Grant Disclosures
The following grant information was disclosed by the authors:
National Forestry and Grassland Administration for Rescues and Breeding of Wild Rare and Endangered Species: 2017, 2018, 2019.
Shenzhen Urban Management Bureau: 201411.
Shenzhen Key Laboratory of Tropical and Subtropical Plant Diversity, Fairy Lake Botanical Garden, Chinese Academy of Sciences: 2017–2019.
South China Botanical Garden: 2017–2019.
Chinese Academy of Sciences.

### Competing Interests
The authors declare that they have no competing interests.

### Author Contributions
- Aimee Caye G. Chang conceived and designed the experiments, performed the experiments, analyzed the data, prepared figures and/or tables, and approved the final draft.
- Qiang Lai performed the experiments, analyzed the data, prepared figures and/or tables, authored or reviewed drafts of the paper, and approved the final draft.

- Tao Chen conceived and designed the experiments, analyzed the data, authored or reviewed drafts of the paper, and approved the final draft.
- Tieyao Tu performed the experiments, analyzed the data, authored or reviewed drafts of the paper, and approved the final draft.
- Yunhua Wang conceived and designed the experiments, authored or reviewed drafts of the paper, and approved the final draft.
- Esperanza Maribel G. Agoo analyzed the data, authored or reviewed drafts of the paper, and approved the final draft.
- Jun Duan conceived and designed the experiments, authored or reviewed drafts of the paper, and approved the final draft.
- Nan Li conceived and designed the experiments, authored or reviewed drafts of the paper, and approved the final draft.

## Data Availability

The complete chloroplast genome of *Microcycas calocoma* is available at GenBank: MN577566.

## Supplemental Information

Supplemental information for this article can be found online at http://dx.doi.org/10.7717/peerj.8305#supplemental-information.

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
