# Peer review of "The complete chloroplast genome of Microcycas calocoma (Miq.) A. DC. (Zamiaceae, Cycadales) and evolution in Cycadales"

_PeerJ, doi:10.7717/peerj.8305_

## Round 0.1 · original submission · Minor Revisions

· Academic Editor

Minor Revisions

Both reviewers and myself think that your contribution is welcomed because the plastid genome of Mycrocycas will contribute to understanding the evolution of Zamiaceae. The paper requires still some changes, particularly on issues regarding classification in Zamiaceae, and position of Microcycas in previous phylogenies. In addition new hypotheses based on your results are needed as well. Below you will find comments by both reviewers that need to be addressed.

Reviewer 1 ·

Basic reporting

The authors provide a valuable contribution by presenting the full chloroplast genome of Microcycas, the only genus that had not previously had a full genome sequenced.

Experimental design

no comment

Validity of the findings

The authors should strengthen their discussion of the results of the phylogeny presented.

The authors repeatedly mention the Cycadales as being divided into three families and present a proposal to classify the order within two families. However, several previously published phylogenies previously supported a division of the Cycadales into two families, and based on the examination of these phylogenies, Christenhusz et al. (2011) already published a formal classification including only the two families Zamiaceae and Cycadaceae. See: http://dx.doi.org/10.11646/phytotaxa.19.1.3 . The authors should therefore not propose this classification, but rather should just state that their topology supports this classification.

No molecular phylogeny supports Stangeria and Bowenia as belonging in the same clade. See 10 examples in figure 2 in Salas-Leiva et al. (2013): https://doi.org/10.1093/aob/mct192 . As with the example above, the authors should mention their results are consistent with all other molecular phylogenies and de-emphasize this finding in the abstract.

The authors should revise the number of extant cycads reported to 355, following The World List of Cycads, a list of accepted cycad names used as a taxonomic reference by the IUCN, CITES, etc...This reference should also be cited in the manuscript. See: https://cycadlist.org/

The authors claim that 40% of species have the possibility of becoming extinct, but this is true of all species, and so the sentence should be clarified. The authors should instead mention the percentage of all cycad species considered threatened by the IUCN Red List of Threatened Species.

The authors attribute the decline of Microcycas calocoma as being due to uneven sex ratios, but this is inconsistent with the following reasons provided by the IUCN Redlist entry for the species: "The plants are affected by habitat destruction (moderate) and over collecting of plants from the wild. Reproductive failure (pollinator extinction) is a concern, although this still needs to be verified." I am not aware of any published research discussing sex ratios in the wild for this species, so the authors should support their assertion with a reference citation or utilize other published data regarding threats to the species, such as the aforementioned IUCN Red List Entry.

Additional comments

The chloroplast genome of Microcycas will be a great resource for future researchers, and the results regarding the genome are well presented. The paper could be greatly improved by improving the interpretation and discussion of the phylogenetic results, as suggested above.

Reviewer 2 ·

Basic reporting

The ideas are well structured, the objective is simple and clear, the English language is good (but there very few little details to solve).

The methods are clear, but few aspects should be better explained ("detailed in "General comments for the author").

Figures 3 and 4 can be summarized in one single figure.

Figure 2 should be better explained in the text (details in "General comments for the author")

The Tables are appropriate in number and information.

Experimental design

The objective is clear: all cycad genera have at least one cpDNA genome reference, except Microcyas. This study aims to fill that knowledge gap.

I think the research is simple but straightforward, making a simple description of the structure and composition of the genome and making phylogenetic comparisons with the other cycad genera.

The technical work is well explained, except for few aspects that are detailed below. Also, the authors will need to be requested to submit the accession number of the genome if the manuscript is accepted.

Validity of the findings

The results are clear and highly descriptive. However, the phylogenetic analyses are especially interesting. The authors apparently confused Stangeria with Bowenia in some part of the discussion (detail below), but I understand that it can be easily corrected.

I think the authors could have been more opened for speculation or more liberal for hypotheses proposal. The results are descriptive, but yet, there is no much discussion about the meaningful of the results. One thing that can improve the quality of this paper is to propose an explanation about the controversial position of Bowenia and Stangeria. Also, it would be nice to appeal better for the distinction of two families in the Cycadales, as many phylogenetic studies don't support the "three families organization". For the experts in cycad phylogenetics, it is clear that Stangeriaceae should not be recognized as a family.

Additional comments

In order of appearance, here are some points that should be considered to improve the quality of the manuscript:


Line 13: Missing comma in: Cycadales – an order of seed plants in subtropical and tropical region[,] comprises...
Line 15: Missing comma in:… ten genera except Microcycas[,] making it an ideal plant group...

Line 34: Please verify “_” in “Wu & Chaw, 201_5"

Lines 80-82: “This ensues because females are more delicate and suffer high mortality rates as they require and expend more resources to reproduce and are more susceptible to die in stressful environments (Field et al., 2013)”
Is this true for cycads in particular? I think it is a possible explanation, but maybe it is too liberal to confirm that this is the cause. Please rephrase to say that “it might be a reason”.

Line 106-107: “Can you please explain why did you choose Bowenia serrulata as a reference genome?

Line 110-111: “Can you please explain how local assembly was done to fill the gaps? Did you use any software? Which one?

Line 138: How many MCMC and burnin did you use in Bayesian inference method for phylogenetic tree construction?

Line 145: Why did Shuffle-LAGAN was used as parameter prior to run the analysis? Also, please change “to running” to “to run”.

Line 179: Cycas debaoensis and C. taitungensis should be in italics.

Line 191: "Sequence variation in cycads” is preferable

Line 192: add space in C. debaoensis

Line 191-199. I am confused with the explanation of Figure 2. The text mentions that purple color indicates conserved sites, and white color indicates sites with indistinct alignment (thus more variable sites). But Figure 2 suggest that purple color indicate protein-coding regions and pink are non-coding regions. I think there is no coherent description between figure and text. Can you please better describe the results shown in Fig. 2?

Lines 206-210. The position of Stangeria is deliberately ommited in this parragraph. But I think it is very interesting and should be explained. The position of Stangeria is not consistent with the concept of the “Neotropical clade”, but this opens a new opportunity to research about the relationship between Stangeria and the neotropical cycads. Please discuss this too.

Sentence 241-242: “The non-functional gene, tufA, was inherited from the charophyte lineage of green algae that once coded for a functional protein elongation factor.”. Please add citation here.

Line 244: Arabidopsis in italics.

Sentence 257-261: This is incorrect. Please check the figures 3 and4 in detail. Actually, Stangeria is joining the Neotropical group, and Bowenia is joining the Encephalartoid group. Please verify whether this confusion affects your ideas for discussion in lines 262-265.
Line 267-274. In addition, the PHYP gene tree produced in Nagalingum et al 2011 shows the same topology as the whole cpDNA tree. I think it is a worthy mention.

Lines 275-288. Can you propose any hypothesis to explain the position of Stangeria as member of the Neotropical group and Bowenia as member of the Encephalartoid group, based in morphological characters? Other previous studies have failed to successfully explain this, and I think this result is a convincing way to finally solve this problem.


In figure 2: Please indicate the extension of LSC, SSC, IRa IRb.

Figures 3 and 4. Topologies and branch lengths are similar. Can you show both results in a single figure? For example, please show only the ML tree, but annotate bootstraps of ML above branches and posterior probability below branches.

---

## Round 0.2 · Minor Revisions

· Academic Editor

Minor Revisions

Please consider a few remaining changes as suggested by the two reviewers - included below and in the annotated pdf file.

Reviewer 1 ·

Basic reporting

Greatly improved manuscript, all my comments are in the attached reviewed pdf

Experimental design

Greatly improved manuscript, all my comments are in the attached reviewed pdf

Validity of the findings

Greatly improved manuscript, all my comments are in the attached reviewed pdf

Additional comments

Greatly improved manuscript, all my comments are in the attached reviewed pdf

Annotated reviews are not available for download in order to protect the identity of reviewers who chose to remain anonymous.

Reviewer 2 ·

Basic reporting

The authors have satisfactorily responded to the reviewers' comments.

As I requested, authors have provided a discussion about the reasons supporting the two-family separation within Cycadales, making emphasis on clarifying the position of the genus Bowenia. Literature is appropiate,

Figures look better, but please be sure to use high-resolution figures for publication (resolution is low in the attached PDF).

GenBank accession is provided.

Experimental design

Methods are well explained, except for one thing:

Authors should provide detailed information about the dataset construction for phylogenetic analyses. Please provide detail instructions to allow for reproducibility. In particular. Please check my questions in "General comments for the author".

Validity of the findings

No comment.

Additional comments

The revision is satisfactory, and I think that the inclusion of the new paragraph in the discussion is good.


In abstract, the first sentence say that "Cycadales is an extant group of seed plants in subtropical and tropical region comprising two families and ten genera". This should mention that they comprise putatively three families and ten genera, since the number of families is something that is discussed a priori based in the results of this research. The same in line 120.


Line 121: "Calonge" should be "Calonje"


About the dataset used for phylogenetic analyses, can you please tell the length of the 81 concatenated genes? Were intergenetic regions excluded? Also, This dataset included all genes from LSC, SSC and IRs? Can you mention it, please?

Lines 426: . "Geographical patterns and distribution of cycads coincide with our phylogenetic analyses using cp genomes but with few exceptions”: Can you please briefly describe the exceptions?


I made a mistake in my previous revision:
I mentioned that the topology of the whole cpDNA tree is similar to that of the PHYP gene shown in Nagalingum et al (2011). This is incorrect. I meant that is similar to the "matK and rbCL" tree shown by Nagalingum et al (2011). Please eliminate the sentence in lines 455-456: (“In addition, the PHYP gene tree produced in Nagalingum et al. (2011) shows the same topology as the whole cpDNA tree”).

---

## Round 0.3 · accepted · Accept

· Academic Editor

Accept

Thank you very much for considering issues and comments raised by the reviewers in the two rounds.